# Underwater Sound Source Localization Based on Passive Time-Reversal Mirror and Ray Theory

**DOI:** 10.3390/s22062420

**Published:** 2022-03-21

**Authors:** Kuan-Wen Liu, Ching-Jer Huang, Gee-Pinn Too, Zong-You Shen, Yung-Da Sun

**Affiliations:** 1Department of Hydraulic and Ocean Engineering, National Cheng Kung University, Tainan 70101, Taiwan; n88071045@gs.ncku.edu.tw; 2Coastal Ocean Monitoring Center, National Cheng Kung University, Tainan 70101, Taiwan; 3Department of Systems and Naval Mechatronic Engineering, National Cheng Kung University, Tainan 70101, Taiwan; z8008070@email.ncku.edu.tw; 4Naval Meteorological and Oceanographic Office, Kaohsiung 81300, Taiwan; g9114072005@mail.cnmoo.mnd.gov.tw

**Keywords:** localization, underwater sound source, passive time-reversal mirror, ray theory, BELLHOP, anchored floating buoy

## Abstract

This study investigates the performance of a passive time-reversal mirror (TRM) combined with acoustic ray theory in localizing underwater sound sources with high frequencies (3–7 kHz). The TRM was installed on a floating buoy and comprised four hydrophones. The ray-tracing code BELLHOP was used to determine the transfer function between a sound source and a field point. The transfer function in the frequency domain obtained from BELLHOP was transformed into the time domain. The pressure field was then obtained by taking the convolution of the transfer function in the time domain with the time-reversed signals that were received by the hydrophones in the TRM. The location with the maximum pressure value was designated as the location of the source. The performance of the proposed methodology for source localization was tested in a towing tank and in the ocean. The aforementioned tests revealed that even when the distances between a source and the TRM were up to 1600 m, the distance deviations between estimated and actual source locations were mostly less than 2 m. Errors originated mainly from inaccurate depth estimation, and the literature indicates that they can be reduced by increasing the number of TRM elements and their apertures.

## 1. Introduction

The localization of underwater acoustic sources is a key research topic because of its application in detecting a range of underwater targets (e.g., fish and submarines). Several methods are used to localize sound sources, including triangulation, beamforming, wave-fingerprint-based techniques (WFPs), and the time-reversal mirror (TRM).

Triangulation is used to estimate the range and bearing of the sound source based on the concept of the time difference of arrival (TDOA) [1,2,3]. In a complex ocean environment, the TDOA has an insufficient resolution for position sensing because of the multipath effects. Beamforming is a signal processing technique used in sensor arrays for directional signal transmission or reception [4]. However, in underwater applications, this method is affected by the problems of inhomogeneous fields and multipath interference, which may distort recorded signals and increase the degradation of beamforming results with increasing signal frequency [5]. Robust adaptive beamforming was developed to enhance the directivity gain, spatial resolution, and suppression of interference and noise [6,7,8]. However, adaptive beamforming is not implemented in shallow water acoustics because of the signal self-cancellation caused by the mismatched signal steering vector and highly-complex calculations used in this method.

WFPs were developed to overcome the multipath effects for object localization using ray-tracing approaches. The basic principle underlying WFPs is that the uniqueness of Green’s functions for different object positions inside a cavity can be exploited for object localization [9,10,11,12]. When a dictionary characterizing the scattering environment is established, the source location can be identified by comparing the measured Green’s function to the established dictionary. Similarly, for objects that do not emit a signal, the object’s scattering contribution is sufficient for object localization. Using the multiple scattered waves to improve localization precision is another prominent feature of WFPs. Eventually, a subwavelength object can be localized by constructing reverberation-coded apertures [12]. WFPs were demonstrated to be promising for indoor localization in complex, dynamic environments [11], but they have not yet been applied to underwater localization.

Acoustic waves that travel through the ocean have low energy dissipation; thus, ocean sound waves can often be time reversed: if $P(\underline{r},t)$ is a solution to the lossless linear wave equation for acoustic pressure, then $P(\underline{r},-t)$ is also a solution. Hence, TRM is also applied to localize underwater acoustic sources. Acoustic time reversal is usually demonstrated with a sound source and a special array of transducers that combine receiver and transmitter functions. This special transducer array is commonly referred to as a time-reversing array (TRA) or TRM [13]. The location that the time-reversed signals emitted from the transducer array focus on is called the retro-focus. Furthermore, the time reversal and the complex conjugate of the pressure field are equivalent to each other. That is, the time reversal of the pressure field in the time domain is equivalent to its phase conjugation in the frequency domain.

The basic principles of acoustic TRMs are similar to those of optical phase-conjugate mirrors [14], which reflect light back towards its source. Jackson and Dowling introduced the concept of phase conjugation to underwater acoustics as a means of determining the route of the sound [15]. Their study established a formal basis for the implementation of acoustic time reversal in underwater applications, which corresponds to the ultrasound research conducted by Fink et al. [16] and Prada et al. [17], and demonstrated the reciprocal phenomena.

The first acoustic time-reversal experiments in the ocean were conducted in Long Island Sound and reported by Parvulescu and Clay [18] and Parvulescu [19]. In the experiments, signals were recorded with a single transducer located 20 nautical miles from a sound source in water that was approximately 1 nautical mile deep. Fink et al., Prada et al., and Fink and Prada [16,17,20] conducted time-reversal experiments for airborne sounds by utilizing a transducer array in the laboratory at ultrasonic frequencies.

Kuperman et al. [21] revised the concept of phase conjugation into the time domain and conducted several experiments in the Mediterranean Sea to demonstrate that the TRM method provides stable retrofocusing for source-array ranges of several kilometers in coastal waters. In the experiments conducted by Kuperman et al. [21], a TRM was implemented by installing a 77 m source-receiver array (SRA) at a water depth of 125 m. The SRA consisted of 20 hydrophones of a cylinder-type sound source with a nominal resonance frequency of 445 Hz. It could receive incident signals from a probe source (PS) and retransmit time-reversed signals to a vertical receiver array (VRA), which was collocated with the PS. The hydrophone of the PS was of the same type used in the SRA. The TRM reduced the signal interference caused by the multipath effect and counteracted the sound reduction in the propagation caused by inhomogeneous flow fields. Based on the experiments of Kuperman et al. [21], Song et al. [22] extended the TRM technique to refocus at ranges that differ from that of the probe source because of a frequency shift at the SRA. The experiments of Kuperman et al. [21] were also emulated by Kim et al. [23] to study the spatial resolution of time-reversal arrays in shallow water. Resolution expressions were derived using an imaging method to describe the achievable focal sizes in various ocean environments. Walker et al. [24] demonstrated that a virtual source array (VSA) could be created by using propagation models or the transfer functions between a TRM and a probe source. A VSA can serve as a remote platform and redirect a focused field to a remote location beyond the VSA that a probe source cannot access. This method is referred to as the TRM–VSA method.

Most TRM experiments were performed using a vertical line array (VLA). Zhang et al. [25] investigated the focus performance of a horizontal time-reversal array, and they reported that a bottom-mounted array provided better performance than those mounted at other depths in shallow water. Normal model modeling was used to explain the aforementioned finding. Zhang et al. [26] introduced virtual time-reversing processing (VTRP) for source localization in shallow water. For VTRP, they used a passive array instead of a source-receiver array. Accordingly, the retransmission of signals in a time-reversed manner was performed by a computer. The parabolic equation method [27,28,29] was used to compute the acoustic field from the source to the VLA. The point source is narrow-banded at a frequency of 170 Hz, and the VLA consisted of 60 elements spanning the water column from 20 to 138 m at a local water depth of 140 m. Their simulation results revealed that relative to matched field processing, VTRP achieved the same localization performance with considerably less CPU time in a range-dependent waveguide. The accuracy of VTRP for source localization was also verified by comparing its results with the experimental data reported by Ginras and Gerstoft [30] and Gerstoft and Ginras [31].

Yu et al. [32] modified the TRM–VSA method proposed by Walker et al. [24] to obtain acoustic images of unburied (proud) and fully buried targets located outside the region between the TRM and the VSA. Their TRM consisted of 32 hydrophones, and their VSA was divided into an upper section and a lower section. The upper section comprised 19 hydrophones, and the lower section comprised 11 hydrophones. The KRAKEN normal mode program [33] was used to calculate the acoustic fields in their simulation examples. On the basis of the VSA-based single time-reversal focusing, Byun et al. [34] developed simultaneous multiple focusing for arbitrarily selected locations. Through numerical simulations, they demonstrated that simultaneous multiple focusing could be achieved and reported that its performance degraded when sound speed mismatches occurred at the boundary between the water column and the sediment layer. Jing et al. [35] proposed a method based on the active detection on virtual time reversal (ADVTR) method for estimating the direction of arrival (DOA) of an underwater target. In contrast to the conventional passive target detection method, which ignores multipath effects, the proposed method incorporates the multipath propagation model. In their study, a sound-propagating algorithm based on the acoustic ray method, namely BELLHOP [36], was used to verify the proposed model.

Conventional underwater acoustic surveillance is performed by deploying connected cables and hydrophones on the sea bed. This method is costly, and the recordings collected through this method are easily distorted. Furthermore, a bottom-deployed system can be easily damaged or destroyed by a bottom trawl during fishing operations. To reduce the fragility of such systems, McDonald et al. [37] developed an autonomous submerged target trip-wire system for detecting and tracking submerged sources. In their study, a 650 m horizontal line array (HLA) with six non-uniformly spaced hydrophones and a 70 m VLA with six uniformly spaced hydrophones were deployed on the seabed. These two array systems ran a matched-field algorithm developed by Bucker [38] and Bucker and Baxley [39]. Their results revealed that the HLA and VLA performed better in terms of horizontal target estimation and depth discrimination, respectively. For real-time detection and tracking, surveillance results were reported through an underwater acoustic modem to a floating Racom buoy, which then transmitted the information to the desired locations.

Instead of a bottom-deployed TRM, an alternative option is to install the TRM on an anchored floating buoy. In addition to being the emplacement of the TRM, the buoy provides a platform for raw data collection and analysis and allows for data transmission through a GPRS (general packet radio service) when the buoy is located close to shore and remote data transmission through a satellite. An anchored floating buoy has been widely used to provide real-time meteorological and oceanographic data [40,41]. Similar to oceanographic data buoys, a buoy with a vertical hydrophone array (which acts as a TRM) may provide information on real-time underwater sound source locations. For this purpose, the present study tested the localization performance of a passive TRM when it was installed on an anchored floating buoy, and the tests were performed in a towing tank and in the ocean.

Four types of models are used to describe sound propagation in the ocean, namely ray theory, spectral method, normal mode, and parabolic equation models [42]. For high frequencies of a few kilohertz or higher, ray theory is the most practical model, whereas the other three are more practical for lower frequencies of less than a kilohertz. In the present study, the sound transmitter (NEPTUNE-TX335, Neptune Sonar Limited, East Yorkshire, United Kingdom) emitted sounds at frequencies ranging from 3 to 7 kHz. Hence, ray theory was applied to describe sound propagation in the ocean. The BELLHOP code [36], which determines the acoustic pressure field based on the ray and beam tracing in the AcTUP (Acoustic Toolbox Use interface and Post processor), was used to determine the pressure field along a ray.

According to the aforementioned literature review, for underwater sound source-localization, most studies used the normal mode model [25,32] and parabolic equation model [26] to describe sound propagation in the ocean. Few studies have examined the performance of a passive TRM based on ray theory in a real ocean environment. This study also tested the localization performance of a passive TRM installed on an anchored buoy. This system can eventually be extended to form an underwater sound monitoring system for reporting real-time 2D source locations. In addition, the procedures for underwater localization based on passive TRM and ray-theory-based model (BELLHOP) are depicted in detail, which is absent in the literature.

This paper is structured as follows. Section 1 provides an introduction explaining how TRM has been applied so far for underwater sound source localization. In Section 2, the TRM method in both the frequency and time domains are described. Section 3 introduces the ray method. Procedures used in this study for underwater sound source localization based on passive TRM and BELLHOP in the AcTUP are also explained. In Section 4, the instrumentation used for localizing the sound source and the laboratory experiments for implementing the developed instrumentation were explained. Section 5 discusses the results of field tests carried out in the offshore region off Small Liuqiu Island. Finally, Section 6 provides the conclusions of this study.

## 2. Time-Reversal Mirror

### 2.1. Time Reversal and Phase Conjugation

The wave equation for the acoustic pressure field with a point sound-source located at $\underline{r_{o}}$ is as follows:(1)$$\nabla^{2}P(\underline{r},t)-\frac{1}{c^{2}}\frac{\partial^{2}P(\underline{r},t)}{\partial t^{2}}=-S\left(t\right)\delta (\underline{r}-\underline{r_{o}})$$
where c is the sound speed, $\underline{r}$ is the location of any field point, S(t) is the strength of the source and has the dimension of N/m, and $\delta (\underline{r}-\underline{r_{o}})$ is the Dirac delta function with its singularity at $\underline{r}=\underline{r_{o}}$. The Fourier transform of Equation (1) is as follows:(2)$$\nabla^{2}p(\underline{r},\omega )+\frac{\omega^{2}}{c^{2}}p(\underline{r},\omega )=-s(\omega )\cdot \delta (\underline{r}-\underline{r_{o}})$$
where ω is the angular frequency and
(3)$$p(\underline{r},\omega )=\int_{-\infty }^{\infty }P(\underline{r},t)\;e^{-i\omega t}dt$$
and the inverse transform of Equation (3) is as follows:(4)$$P(\underline{r},t)=\frac{1}{2\pi }\int_{-\infty }^{\infty }p(\underline{r},\omega )\;e^{i\omega t}d\omega =\int_{-\infty }^{\infty }p(\underline{r},\omega )\;e^{i\omega t}df$$

Similarly,
(5)$$s(\omega )=\int_{-\infty }^{\infty }S(t)\;e^{-i\omega t}dt$$

From Equation (2), it is known that a time-harmonic, or monochromatic, point sound-source generates spherical waves with an acoustic pressure field that can be expressed as follows:(6)$$P(r,t)=p(r,\omega )\;e^{i\;\omega \;t}=s(\omega )\cdot G(\underline{r},\underline{r_{o}},\omega )e^{i\;\omega \;t}=s\cdot \frac{e^{i(-k\;r+\varphi )}}{4\pi r}e^{i\;\omega \;t}=s\cdot \frac{e^{i(\omega t-k\;r+\varphi )}}{4\pi r}$$
where k is the wavenumber (k=ω/c), r is the distance between $\underline{r}$ and $\underline{r_{o}}$, φ is the phase angle of the source, and $G(\underline{r},\underline{r_{o}},\omega )$ is Green’s function for the Helmholtz equation Equation (2). The waves expressed in Equation (6) denote the waves that propagate away from the sound source.

When time is reversed (i.e., t is replaced by −t), the pressure field becomes
(7)$$P(r,-t)=p(r,\omega )\;e^{-i\;\omega \;t}=s\cdot \frac{e^{-i(\omega t+k\;r-\varphi )}}{4\pi r}$$

The pressure field expressed in Equation (7) denotes the waves that converge toward the sound source.

Time-reversed waves can also be constructed by simultaneously changing the signs of kr and φ in Equation (6) while leaving the sign of t unchanged. Accordingly, it yields a pressure field that can be expressed as follows:(8)$$s\cdot \frac{e^{i(k\;r-\varphi )}}{4\pi r}\cdot e^{i\omega t}=p^{*}(r,\omega )\cdot e^{i\;\omega t}=s\cdot \frac{e^{i(\omega \;t+k\;r-\varphi )}}{4\pi r}$$

The pressure expressed in Equation (8) also denotes the waves that converge toward the sound source. From Equations (7) and (8),
(9)Re[P(r,−t)]=Re[p(r,ω)e−iωt]=s4πrcos(ωt+kr−φ)=Re[p*(r,ω)eiωt]

The aforementioned derivation was proposed by Dowling and Song [13] to illustrate that for pure time-harmonic signals, the time reversal and complex conjugation (denoted by a superscripted *) of p(r) have the same meaning. Because broadband signals can be constructed through a Fourier superposition of single-frequency signals, time reversal is equivalent to complex conjugation in the frequency domain not only for single-frequency signals but also for broadband signals.

### 2.2. Implementation of TRM

The present study involved the localization of an underwater acoustic source through the use of a vertical hydrophone array, which acted as a TRM. The acoustical signals emitted from a probe source (PS) were recorded by the hydrophone array, after which the recorded sounds were subjected to time reversal and then back-propagated.

As highlighted in the previous section, time reversal is equivalent to phase conjugation in the frequency domain. Accordingly, the pressure field $p_{pc}$ due to M discrete sources in a TRM can be derived from Equation (6) and expressed as follows [15,43]:(10)$$p_{pc}(r,z,\omega )=\sum_{j=1}^{M}G_{\omega }(r,z;R,z_{j})\cdot p_{\omega }^{*}(R,z_{j};0,z_{PS})$$
where M denotes the number of hydrophones in the receiver array (TRM), $p_{\omega }^{*}(R,z_{j};0,z_{PS})$ are the phase conjugated frequency-dependent pressure values recorded at each element of the array at a range of R and depth of $z_{j}$ when the probe source is at a depth of $z_{PS}$, and $G_{\omega }(r,z;R,z_{j})$ denotes the Green’s function for each array element at the field point with a range of r and depth of z around the probe source location. $G_{\omega }(r,z;R,z_{j})$ can also be interpreted as the transfer function between the discrete sources in the TRM and a field point. This transfer function incorporates scattering, multipath, and waveguiding effects. In the present study, the sound transmitter (NEPTUNE-TX335) emitted high-frequency sounds of 3–7 kHz. Therefore, ray theory was applied to describe sound propagation in the ocean and $G_{\omega }(r,z;R,z_{j})$ was determined using the BELLHOP code in the AcTUP, which is described in Section 3.2.

### 2.3. TRM in the Time Domain

In the previous section, the TRM is depicted in the frequency domain. This section provides a similar depiction but in the time domain. Assume that there is a probe sound source that emits time-series signal s(t), refer to Figure 1a, and the sound signal received by a single hydrophone in the TRM is r (t), refer to Figure 1b, then based on Equation (6) and the convolution theorem, which states that multiplication in the frequency domain is equivalent to convolution in the time domain, it yields
(11)r(t)=g(t)⊗s(t)
where g(t) can be interpreted as the transfer function between the source signal and the received signal or the impulse response function (IRF), and ⊗ denotes the convolution. Based on the definition of convolution, Equation (11) can be expressed as follows:(12)$$r\;(t)=g(t)⊗s(t)=\int_{-\infty }^{\infty }g(\tau )s(t-\tau )d\tau$$

When the received signal r(t) is time reversed, Equation (11) becomes:(13)r(−t)=g(−t)⊗s(−t)

Let the reversed signal r (−t) is emitted from the receiver, ref to Figure 1c, then the signal received at the location of the probe source, z (t), refer to Figure 1d, is
(14)z(t) =g(t)⊗r(−t)=g(t)⊗g(−t)⊗s(−t)

If the received signal z (t) is time reversed, Equation (14) becomes
(15)z (−t)=g(−t)⊗g(t)⊗s(t)=const ⋅ s(t)

Equation (15) indicates that the time series of z(−t) are similar to the original sound source signal s(t), except a multiplication of a constant. This demonstrates that time reversal mirror can reconstruct the wave forms and time series of the original signal, except the energy value. The accuracy of the reconstructed source signals mainly depends on the value of impulse response function g(t). Notably, the reconstruction of the wave forms in Equation (15) was obtained based on the assumption that the transfer function for the sound propagating from the source to the receiver is identical to that for the back propagation. In actual ocean environments, this may not be true and will affect the reconstruction accuracy.

In the case when there are several hydrophones in the TRM, the corresponding forms for Equations (11), (13)–(15) are:(16)$$r_{m}(t)=g_{m}(t)⊗s(t)$$
(17)$$r_{m}(-t)=g_{m}(-t)⊗s(-t)$$
(18)$$z_{m}(t)=g_{m}(t)⊗r_{m}(-t)$$
(19)$$z_{m}(-t)=g_{m}(-t)⊗g_{m}(t)⊗s(t)=const\;\cdot \;s(t)$$
where the subscript m=1,2,3,…,M. The time-reversed signals at the original source location from the multiple hydrophones can be obtained by adding the individual signals $z_{m}(-t)$ together:(20)$$z_{sum}(t)=\sum_{m=1}^{M}z_{m}(t)=\sum_{m=1}^{M}g_{m}(t)⊗r_{m}(-t)$$

Or
(21)$$z_{sum}(-t)=\sum_{m=1}^{M}z_{m}(-t)=\sum_{m=1}^{M}g_{m}(-t)⊗g_{m}(t)⊗s(t)$$

Equation (20) is equivalent to Equation (10), except that in Equation (10), the pressure is determined at any field point rather than at the source location.

### 2.4. Practical Implementation of TRM

In actual practice, a TRM can be implemented actively or passively. For the active method, the sound signals received by the hydrophones in a receiver array (i.e., a TRM) are reversed and emitted for localization; for the passive method, reversed sound signals are used as sound sources, algorithms based on Equation (10) or Equation (20) are developed to determine the sound pressure field in a selected area, and the location with the highest sound pressure is designated as the position of the targeted source. In the present study, a passive TRM was used for the localization of underwater sound targets.

## 3. Ray Methods

In order to describe the propagation of high-frequency sounds in the ocean, ray theory is the most practical model. This section briefly describes how the equations used in ray theory were derived from the Helmholtz equation. Refer to Jensen et al. [27] for a detailed discussion of this topic.

### 3.1. Ray Equations

The Helmholtz equation for the acoustic pressure field with a sound source of unit strength located at $\underline{r_{o}}$ is
(22)$$\nabla^{2}p+\frac{\omega^{2}}{c^{2}(\underline{r})}p=-\delta (\underline{r}-\underline{r_{o}})$$
where $c(\underline{r})$ is the sound speed in Cartesian coordinates $\underline{r}=(x,y,z)$. In order to obtain the ray equations, the solution to the Helmholtz equation is assumed to have the following form [44]:(23)$$p(\underline{r})=e^{i\omega \tau (\underline{r})}\sum_{j=0}^{\infty }\frac{A_{j}(\underline{r})}{{\left(i\omega \right)}^{j}}$$
where $\tau (\underline{r})$ denotes the level curves of wavefront and is also referred to as eikonal. Based on this solution, it is obvious that in order t to obtain the pressure field for high wavenumber cases, the terms in Equation (23) with j≥1 can be ignored. Accordingly,
(24)$$p(\underline{r})=A_{o}(\underline{r})\cdot e^{i\omega \tau (\underline{r})}$$

The pressure field given in Equation (24) is valid for a sound source of unit strength. Hence, $p(\underline{r})$ can be interpreted as the transfer function between the source and a field point. It differs from Green’s function (given in Equation (6)) in that the latter is valid only when the sound velocity $c(\underline{r})$ in Equation (22) is a constant.

Substituting Equation (23) into Equation (22) and equating terms of the same order in ω, the following equations for the functions $\tau (\underline{r})$ and $A_{j}(\underline{r})$ can be obtained:(25)$$O(\omega^{2}):\;{\left|\nabla \tau \right|}^{2}=\frac{1}{c{\left(\underline{r}\right)}^{2}}$$
(26)$$O(\omega ):\;2\nabla \tau \cdot \nabla A_{0}+(\nabla^{2}\tau )A_{0}=0$$
(27)$$O(\omega^{1-j}):\;2\nabla \tau \cdot \nabla A_{j}+(\nabla^{2}\tau )A_{j}=-\nabla^{2}A_{j-1},\;j=1,2,\ldots$$

The $O(\omega^{2})$ equation for $\tau (\underline{r})$ is known as the eikonal equation. The remaining equations for $A_{j}(\underline{r})$ are known as the transport equations. Equations (25) and (26) can be solved to obtain $\tau (\underline{r})$ and $A_{o}(\underline{r})$ in Equation (24), respectively.

In the solution of the eikonal equation, a family of curves (rays) perpendicular to the level curves (wavefronts) of $\tau (\underline{r})$ is introduced. This family of rays defines a new coordinate system called ray coordinates. Since ∇τ is a vector perpendicular to the wavefronts, one can define the ray trajectory $\underline{r}(s)$ by the following ray equation:(28)$$\frac{d\underline{r}}{ds}=c\nabla \tau$$

Here s denotes the distance along the ray. In the coordinate system of the rays, the eikonal equation can be rewritten as
(29)$$\frac{d\tau }{ds}=\frac{1}{c}$$

Equation (29) can be solved to yield
(30)$$\tau (s)=\tau (0)+\int_{0}^{s}\frac{1}{c(s\prime )}\;ds\prime$$

The introduction of the ray coordinates provides an easy way to determine the travel times and amplitudes along each ray, which then yields the pressure field along each ray. In practice, one needs to interpolate the values from the ray grids to the rectangular grids. This transformation can be made by constructing a hat-shaped beam around each ray [45]. The hat-shaped function decreases linearly from $A_{o}(s)$ on the central ray of the beam to zero on either side. Thus, the pressure field for the beam is given by
(31)$$p(s,n)=A_{o}(s)\cdot \phi (s,n)\cdot e^{i\omega \tau (s)}$$
where $A_{o}(s)$ is the amplitude with each ray, ϕ(s,n) is the hat-shaped function, and n is the normal distance from the ray.

### 3.2. AcTUP

The AcTUP is an acoustic toolbox for underwater acoustics, and its original code was written by Michael Porter from Heat, Light, and Sound Research [33,36,45]. The AcTUP V2.2ℓ toolbox was released [46,47]. This is an open graphical user interface (GUI) written using MATLAB, and it provides access to programs that can perform acoustic field calculations by utilizing normal mode (KRAKEN), wavenumber integration (FIELD), ray and beam tracing (BELLHOP), and parabolic equation (RAMGEO) methods. In the BELLHOP model, the free surface is treated as a pressure-released boundary. Furthermore, the BOUNCE program can be used to determine the plane wave reflection coefficient for a layered seabed. In the present study, the BELLHOP code was used to determine the pressure field along the ray using Equation (31), and BOUNCE was used to determine the wave reflection from the seabed. BELLHOP is a finite-element ray-tracing algorithm [45]. Ray tracing is applicable to range-dependent problems, and it tends to be more accurate for high-frequency sound sources. Therefore, ray tracing is suitable for high-frequency, broadband sound source problems in a range-dependent environment [36].

### 3.3. Procedures for Underwater Localization Based on Passive TRM and BELLHOP

The sound transmitter (NEPTUNE-TX335) emits a series of sound pulses
s(t)
at the frequencies of 3, 4, 5, 6, and 7 kHz;The emitted sounds are received by the four hydrophones in the TRM. The received sound signals are denoted as
$r_{m}(t)\;(m=1,2,3,4)$;The time series of
$r_{m}(t)$
is reversed to obtain
$r_{m}(-t)$;The pressure field for the sound beam (i.e.,
p(s,n)
in Equation (31)) is calculated using the
BELLHOP and BOUNCE algorithms
in the AcTUP.
p(s,n)
is used as the transfer function or the impulse response function (IRF; i.e.,
$G_{\omega }(r,z;R,z_{j})$
in Equation (10));The transfer function in the frequency domain
$G_{\omega }(r,z;R,z_{j})$
with
j=1,2,3,4
is transformed into the transfer function in the time domain,
$g_{m}(t)$, by using MATLAB;Two-dimensional grids are generated in the domain of interest by using MATLAB. In the present study, the grid sizes for both the depth and range axes were 0.01 m, expressed as
Δr=Δz=0.01 m;Through MATLAB,
$r_{m}\;(-t)$
and
$g_{m}(t)$
are convoluted on the 2D grids to obtain
$z_{m}(t)$
in Equation (18);The
$z_{m}(t)$
of the hydrophones on the TRM are summarized to obtain
$z_{sum}(t)$
in Equation (20), and the time-series data of
$z_{sum}(t)$
within a 1 s period are cumulated to obtain a corresponding time-integrated value for
$z_{sum}(t)$, and denoted as
$z_{SUM}$;The location with the maximum value of
$z_{SUM}$
is assumed to be the location of the underwater acoustic source.

## 4. Instrumentation and Laboratory Experiments

As was highlighted in the Introduction section, this study proposed the installation of a TRM on an anchored floating buoy for the purpose of conducting underwater acoustic surveillance. The buoy provides a platform for data collection and analysis and for data transmission through a GPRS or satellite. Theoretically, similar to the ocean monitoring data buoy [40,41], the proposed underwater sound monitoring system (USMS) can provide real-time source locations. In the present study, tests were conducted in a towing tank and in the ocean to assess the performance of a passive TRM installed on an anchored floating buoy for underwater source localization. Because the data analysis was performed after the experiments were complete, real-time source locations could not be reported through a GPRS or satellite. The NEPTUNE-TX335 was used to emit high-frequency sounds (3 to 7 kHz). The NEPTUNE-TX335 used to emit high-frequency sounds (3–7 kHz) is of cylinder type, and the emitted sounds are in the plane wave form. Consequently, ray theory was used to obtain the acoustic pressure field.

The proposed USMS design comprises an anchored floating data buoy; a power supply; a vertical hydrophone array that acts as a TRM; and a computer that contains a data acquisition module, a fourth-generation (4G) network module, and a Global Positioning System (GPS) module. This design allows for components to be upgraded. In areas with poor 4G network performance, a satellite can be used to transmit collected data to a desired remote location.

### 4.1. Main Component Features of the USMS

The buoy used in the present study was developed through a collaboration with the R&D and System-Integration teams of the Coastal Ocean Monitoring Center, National Cheng Kung University (NCKU). The buoy hull was designed to carry a USMS and two battery packs to provide power for the equipment. For long-term monitoring, additional solar panels can be installed to charge the batteries. The hydrophone array with the preamplifier was attached to the buoy and straightened using a weighted pack when the tests were carried out in the towing tank. In the field tests, the hydrophone array was attached to the mooring line of the buoy. The cargo space in the buoy hull had a depth of 40 cm and a diameter of 12 cm to accommodate the computer, data acquisition module, and batteries. In order to minimize the clicking noise caused by the shifting of equipment, each piece of equipment was secured to a steel shelf in the hull. The hatch on the top of the buoy hull was sealed to waterproof the cargo space. In order to stabilize the floating motion of the buoy, a foam cylinder with a diameter of 60 cm and a height of 20 cm was wrapped around the hull. A rod measuring 45 cm in length was mounted on the top of the hull; a flag was hung on the rod to indicate the position of the buoy, and an antenna was attached to the rod to enable the operation of the GPS system.

A single-board computer (called UP-board) manufactured by AAEON Technology Inc. (New Taipei City, Taiwan) was used to meet the specific signal acquisition needs of the present study. The computer had a sufficiently fast processor (2.5 GHz) with 256 GB memory, one Universal Serial Bus (USB) 3.0 port for the data acquisition module, and small dimensions that allowed for installation in the hull of the buoy. Additionally, the computer had to operate reliably on the sea surface and in high-temperature environments. It also had to store large amounts of data.

A USB-2405 data acquisition module manufactured by ADLINK Technology Inc. (Taipei, Taiwan) was used. This 24-bit high-performance dynamic signal acquisition module is USB bus-powered and equipped with the following features: BNC connectors and removable spring terminals for device connectivity, four analog input channels for simultaneous sampling at 128 kS/s per channel, a software-selectable alternating-current or direct-current coupling input configuration, and a built-in high-precision 2 mA excitation current that enables measurements using integrated electronic piezoelectric sensors. Each input channel was connected to the hydrophones in this manner. The onboard 24-bit Sigma-Delta ADC supports anti-aliasing filtering, which suppresses modulator and out-of-band signal noise. It provides a usable signal bandwidth at the Nyquist rate, which makes it ideal for performing high-dynamic-range signal measurements in acoustic applications. The data acquisition module is attached to the computer via a USB plug, and it starts to operate upon being connected. During the experiment, the computer was connected to a technician’s cellphone or notebook to enable the remote control of data acquisition.

The TRM consists of four HTI-94-SSQ hydrophones (High Tech Inc., Long Beach, MS, USA). Because the measured sound signals were usually too weak to be directly read, a charge amplifier had to be connected between each hydrophone and the data acquisition module. The HTI-94-SSQ hydrophones used in the tests were each equipped with a preamplifier. The hydrophones were calibrated to ensure that the measurement results matched the results generated by a reference device, namely a B&K 8104 hydrophone (Brüel and Kjær, Virum, Denmark), which was previously calibrated using the B&K hydrophone calibrator (Type 4229).

### 4.2. Laboratory Experiments

After the completion of system integration, laboratory experiments were performed to test the stability of the system, the correctness of the signals, and the accuracy of the algorithm used for sound source localization. The laboratory experiments were conducted on 21 July 2018, in the towing tank of the Department of Systems and Naval Mechatronic Engineering, NCKU. The towing tank has a length of 165 m, a width of 8 m, and a depth of 4 m. The experiments were conducted at a water depth of 3.5 m. The NEPTUNE-TX335 transducer was used as the sound transmitter, and it was supported by a fixed steel frame and placed at a distance of 20 m from the leading edge of the tank at a water depth of 2.75 m. Figure 2 illustrates the layout of the laboratory tests conducted in the towing tank. The floating buoy with the hydrophone array was positioned at distances of 4, 10, 20, 40, 60, and 80 m from the transmitter. At each distance, the array recorded the sound emitted from the transmitter, and the sampling rate for measurements was 40 kHz. The sounds were emitted for 1 min (with a period of 0.5 s and a signal duration of 0.2 s) at the frequencies of 3, 4, 5, 6, and 7 kHz.

During the experiments, the water temperature was $26{\;}^{o}C$, the corresponding density $\rho_{w}$ was 997.2 ${\mathrm{kg}/m}^{3}$, and the sound speed was 1485.3 m/s, which is determined using the following formula:(32)$$c=\sqrt{\frac{E_{v}}{\rho_{w}}}$$
where $E_{v}$ denotes the bulk modulus of water and is $2.2\times 10^{9}\;{N/m}^{2}$ when the water temperature is $26{\;}^{o}C$. The speed of sound was assumed to be homogeneous throughout the towing tank. The attenuation of sound in water was ignored in the BELLHOP algorithm used in the present study.

### 4.3. Results and Discussion

As highlighted in Section 2, in the present study, the TRM provided only 2D sound source localization; hence, range (r) and depth (z) are provided, but not bearing. Accordingly, the domain where the sound pressure was determined was identical to that presented in Figure 2; it had a horizontal range of 65 m and vertical range of 3.5 m, and the computational resolution was 0.01 m for both the horizontal and vertical directions (e.g., Δr=Δz=0.01 m).

Figure 3 presents the typical sound signals received by the second hydrophone ($h_{2}=1.5\;m$) in the TRM. In Figure 3, the distance between the TRM and the transducer is 80 m, and the frequency of the sound source is 3 kHz. Figure 3 indicates that in addition to the sound source signals, the reflected sound signals from the water surface and the side walls of the tank were also detected. Figure 4 presents the typical sound signals received by all hydrophones in the TRM. Because the difference in the distance between the sound source and each hydrophone was negligibly small, the received signals were superimposed over each other and difficult to distinguish. Figure 5 shows the sound pressure at the retrofocused location in the region close to the original sound source location when the sound source was located 80 m from the TRM at a water depth of 2.75 m, namely $r_{o}=80\;m$ and $z_{o}=2.75\;m$. The pressure shown in Figure 5 was determined from Equation (20). The transfer function in Equation (20) has a unit of $m^{-1}$ and $r_{m}(-t)$ was obtained from the time reversal of $r_{m}(t)$, which was measured by the hydrophone on TRM and had a unit of Volt (V). Hence, the pressure in Figure 5 has a unit of V/m. Figure 5 indicates that the estimated sound source locates at $r_{oe}=80.89\;\;m$ and $z_{oe}=2.54\;\;m$. The distance between the estimated and actual source locations, $d_{o1}=\sqrt{{\left(r_{oe}-r_{o}\right)}^{2}+{\left(z_{oe}-z_{o}\right)}^{2}}$, was 0.91 m (Table 1).

Table 1 summarizes the results of the estimated source locations at various source locations (4–80 m) and sound frequencies (3–7 kHz) as obtained from the experiments conducted in the towing tank. Notably, the distances between estimated and actual sound source locations (denoted as distance deviation $d_{01}$) were mostly less than 2 m. Large deviations originated mainly from depth errors. The range errors in all studied cases were less than 1 m, even when the sound source was 80 m from the TRM. However, the depth errors were considerably large.

Table 1 also presents the average absolute errors for the range, depth, and distance between the actual and estimated source locations. The average of the values obtained at various sound frequencies was computed. Notably, Table 1 indicates that the percentage of the range error decreased when the range increased; however, the percentage of the depth error and the average value of $d_{01}$ did not change in response to changes in the source location or sound frequency. The results obtained from the towing tank experiments revealed that with only four hydrophones and an aperture (i.e., the interval between hydrophones) of 0.5 m in the TRM, relative to the results of previous experiments in which more than 20 hydrophones were usually used [21,26,32], the present passive TRM configuration combined with the AcTUP allowed for reasonably accurate source locations to be obtained. Through their numerical simulations, Sun [48] and Chen [49] demonstrated that an increase in the number of TRM elements and TRM’s aperture resulted in higher sound pressure at the retro-focused location. Accordingly, having more TRM elements and a larger aperture increases the accuracy of source localization. The performance of the proposed source localization algorithm required further field testing, which is described in the subsequent section.

## 5. Field Tests

After the localization performance of the passive TRM installed on a floating buoy was verified in a towing tank, field tests were conducted to examine the performance in actual ocean environments. On 25 April 2020, field tests were conducted in the offshore region off Small Liuqiu Island, Taiwan. The test location (22°19′83″ N, 120°23′02″ E) was 1 km from the south-eastern beach of the island. The local water depth was 77 m. During the tests, four bottles of seawater were collected at water depths of 1, 1.5, 2, and 2.5 m to measure temperature and salinity levels. At the depths of 1, 1.5, 2, and 2.5 m, the measured temperatures were 28.5, 27.2, 27.0, and 26 °C, respectively; the measured salinity levels were 35.5, 35.8, 35.6, and 35.7 psu; and the associated sound speed was 1543, 1540, 1540, and 1538 m/s, respectively. The average seawater density ($\rho_{sw}$) was 1023 kg/m^3^. At the area where the tests were performed, the sound speed profile from 2.5 to 77 m was assumed to be uniform (Figure 6).

The sound speed in the seawater was calculated using the following formula [50]:(33)$$\begin{array}{l} c=1449.2(m/s)+4.6T-0.055T^{2}+0.00029T^{3}\\ \;\;\;+(1.34-0.01T)(S-35)+0.016z \end{array}$$
where T is the temperature in degrees Celsius, S is the salinity in ppt, and z is the water depth in meters. Although the salinity in Equation (33) is expressed in ppt, because the numerical difference between psu and ppt is small, the salinity expressed in psu was used to determine the sound velocity from Equation (33).

The density of seawater at 1 atm (denoted as $\rho_{sw}$) is calculated using the following equation [51]:(34)$$\rho_{sw}=\rho_{w}+AS+BS^{1.5}+CS^{2}$$
where $\rho_{w}$ is the density of pure water (no salinity); *S* is the salinity of seawater in ppt; and *A*, *B* and *C* are coefficients, which values depend on the temperature.

The rigidity of sediment is usually substantially less than that of a solid, and accordingly, bottom sediment is treated as a fluid. The sediment layer at the offshore region off Small Liuqiu Island is several meters deep. In this study, it was assumed to be 10 m deep. A field survey revealed that the density of the sediment layer was 1750 kg/m^3^ and that the sound speed in the sediment was 1563 m/s [52,53]. The sound attenuations in both the water column and sediment layer were ignored for the field tests conducted in the offshore region off Small Liuqiu Island. Hence, both $\alpha_{w}$ and $\alpha_{s}$ were set to zero. Figure 7 presents the parameters and geometry of the TRM experiments that were conducted in the offshore region off Small Liuqiu Island. The sound transmitter emitted a series of sound pulses every 0.7 s at the frequencies of 3, 4, 5, 6, and 7 kHz. The duration of each pulse was 0.3 s. The sound transmitter was installed on the side of a boat, and it was positioned at a water depth of 2.5 m.

Figure 8 presents the time series of signals received by the third hydrophone in the TRM, which was 550 m from the sound source with a frequency of 3 kHz. Figure 9 presents the sound signals within a period of $2\times 10^{-3}\;s$ that were received by the third hydrophone in the TRM, and these results demonstrated that the sound frequency was 3 kHz. Figure 10 presents the calculated sound pressure levels near the sound source location, which was located 550 m away from the TRM at a depth of 2.5 m. Figure 10 reveals that the estimated sound source location was $r_{oe}=548.6\;\;m$ and $z_{oe}=2.45\;\;m$. The distance between the actual and estimated source locations ($d_{01}$) was 1.36 m (Table 2). Figure 11 and Figure 12 present the results that correspond to those presented in Figure 8 and Figure 10 when the aforementioned distance increased to 1600 m. Figure 12 indicates that the estimated source location is $r_{oe}=1601.9\;\;m$ and $z_{oe}=2.60\;\;m$, and the distance between the actual and estimated sound source locations ($d_{01}$) was 1.90 m.

Table 2 summarizes the actual and estimated source locations and the distance between the actual and estimated locations as obtained from the Small Liuqiu Island tests conducted at different source locations and sound frequencies. The results in Table 2 reveal that the range errors were usually negligibly small; for example, when $r_{o}=550\;m$, the average error was 1.72 m or 0.31%. By contrast, the depth errors were considerably larger, and the average error was 0.466 m or 18.64%. The average $d_{01}$ value was 1.808 m. When $r_{o}=1100\;m$, the average range error was 2.64 m or 0.24%, the average depth error was 0.468 m or 18.72%, and the average $d_{01}$ value was 2.722 m. Similarly, when $r_{o}=1600\;m$, the average range error was 0.70 m or 0.04%, the average depth error was 0.742 m or 29.68%, and the average $d_{01}$ value was 1.864 m. Notably, Table 2 reveals that the characteristics of the estimated source locations obtained from the Small Liuqiu Island tests are consistent with those obtained from the towing tank experiments. The main characteristic is that the implementation of a TRM with a VLA of four hydrophones allows for the accurate estimation of the source range; however, this model also results in unsatisfactory depth accuracy. The poor accuracy of source depth could be improved if more hydrophones with a larger aperture were installed in the line array to extend its depth range.

Several reasons could explain the source depth errors in the results obtained from the field tests. The TRM was installed on a floating buoy, which moved up and down on the water surface. The variation in the vertical position of the TRM due to the effect of surface waves and the effects of the local currents in bending the VLA are factors that increased the difficulty of obtaining an accurate position of the hydrophones. During the Small Liuqiu Island tests, a small significant wave height of 0.38 m was detected by a nearby data buoy. Hence, the effects of surface waves on location error should be minimal. Furthermore, in the simulation that was performed using AcTUP, the seabed was assumed to be flat, but this assumption does not reflect actual seabed conditions.

A comparison of the results presented in Figure 8 and Figure 11 also revealed that the signals received by the TRM elements decreased when the distance between the sound source and TRM increased. This finding indicates that a further increase in this distance may result in the received signals becoming indistinguishable from environmental noises and limit the localization capability of the proposed TRM configuration that incorporates BELLHOP code.

This study used the ray-tracing code BELLHOP to determine the transfer function between a sound source and a field point. As indicated in Section 3, the ray-tracing method is based on linear wave propagation and does not consider sound absorption and dispersion in water or diffraction and scattering when sound waves encounter an object. Scattering also occurs when sounds are bounced by a rough boundary. Sound absorption in seawater is insignificant below 1 kHz, and even at 10 kHz, the attenuation factor α is 0.60 dB/km (for water at 20 °C under 1 atm) based on the attenuation formula of Fisher and Simmons [54]. The corresponding value at 5 kHz is 0.24 dB/km. Therefore, the transmission loss of sound caused by absorption is insignificant in the frequency range studied herein. Acoustic dispersion is the phenomenon of a sound wave separating into its component frequencies because of the dependence of sound propagation speed on signal frequency. Therefore, the effect of acoustic dispersion is relatively insignificant for narrow-band signals and can be neglected for the single-frequency signals used in this study unless nonlinear phenomena occur such that higher harmonics are generated. However, for broadband signals propagating in a deep ocean, because of the complexity of the ocean channel, dispersive effects should be considered [55]. When a sound wave encounters an obstacle, a scattered wave is generated and spreads out from the obstacle in all directions. The scattered wave interferes with the incoming wave and results in the change of wave direction. Diffraction refers to the case when the scattering object is large compared with the wavelength of the scattered sound, and scattering refers to the case when the obstacle is very small compared with the wavelength [56,57,58].

These factors reduce the intensity of the received sounds and prompt substantial multipath effects that result in localization errors. However, when more elements are added to the time-reversal array to span a wider space, the fraction of the original wave front that is time-reversed increases, such that the diminishing intensity of the received sounds and the multi-path effects can be compensated for and the effectiveness of the retrofocusing can be improved. This is why the localization accuracy can be improved by increasing the number of TRM elements and their apertures, as demonstrated by Sun and Chen [48,49] through numerical simulations.

Comparing the performance of the proposed TRM with available sonar systems for the same localization purpose (e.g., range and depth) would be beneficial. However, this is beyond the scope of this study. For detailed information on the localization performance of various sonar systems, refer to Hodges [59].

Notably, because the source locations were not determined right after the completion of the tests, the source locations could not be reported through a GPRS or satellite to the desired land locations. This limitation of the present study will be addressed in our future research. Furthermore, the present study only determined the range and depth of the sound source. The bearing of the source was not determined.

## 6. Conclusions

In order to reduce the fragility of a bottom-deployed TRM, an alternative option is to install the TRM on an anchored floating buoy. The present study investigated the performance of a passive TRM that was installed on a buoy for underwater sound source localization. The sounds emitted by the NEPTUNE-TX335 transducer had high frequencies ranging from 3 to 7 kHz. Accordingly, BELLHOP code, which was developed based on acoustic ray theory, was adopted to determine the transfer function in the frequency domain between the probe source and the TRM. The TRM comprised a VLA with four hydrophones and an aperture of 0.5 m.

The performance of the proposed TRM combined BELLHOP code for source localization was examined by conducting laboratory experiments in a towing tank and field tests in offshore regions off Small Liuqiu Island at a local water depth of 77 m.

These test results revealed that in most cases, the distance between the estimated and actual source locations was less than 2 m even when the distance between the sound source and TRM was up to 1600 m. Errors originated mainly from inaccurate depth estimation, and they can be reduced by increasing the numbers of TRM elements and the size of the aperture.

The present study suggests that implementing a design comprising an anchored floating buoy with a VLA that acts as a TRM, a power supply, a computer with a data acquisition module, a GPRS module for real time data transmission, and an algorithm for estimating source locations, can yield a USMS for reporting real-time 2D underwater sound source locations.

## Figures and Tables

**Figure 1 sensors-22-02420-f001:**
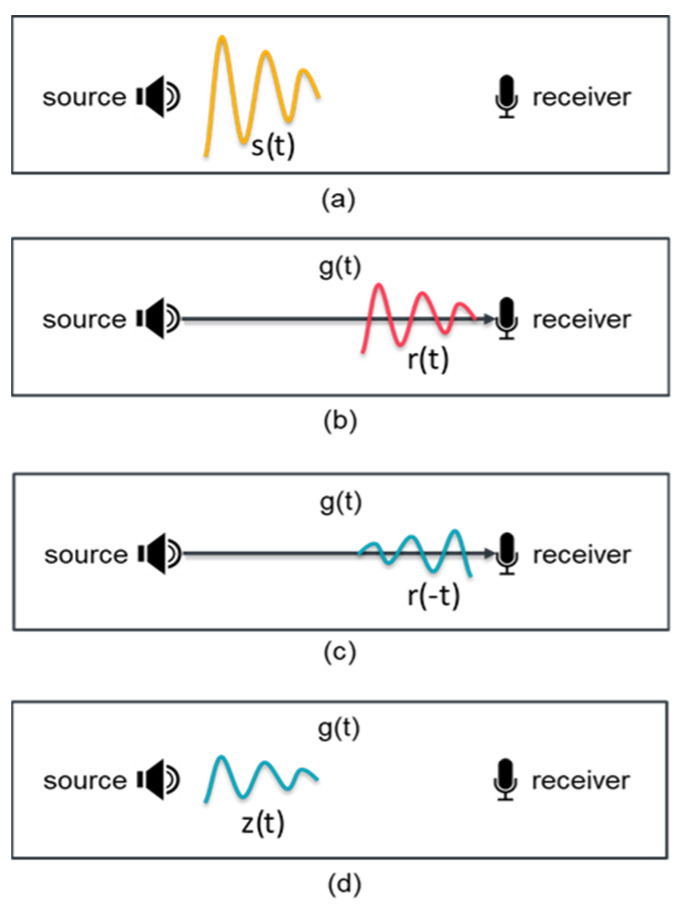
A time-reversal mirror (TRM) in the time domain. (**a**) Sound source signal *s*(*t*), (**b**) received sound signal *r*(*t*), (**c**) time-reversed signal of *r*(*t*), and (**d**) signal received at the location of the probe source.

**Figure 2 sensors-22-02420-f002:**
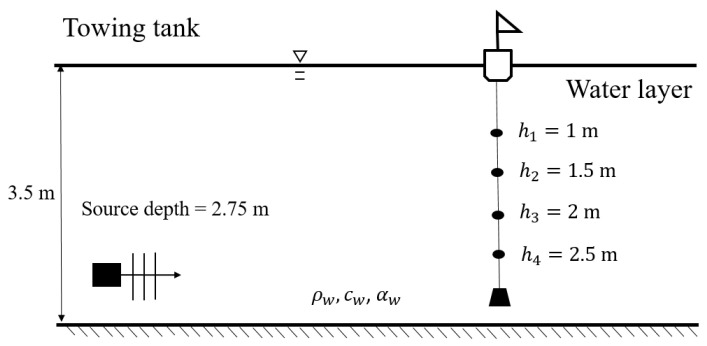
Layout of laboratory tests in the towing tank. The interval between two adjacent hydrophones is 0.5 m, the interval between the topmost hydrophone and the water surface is 1 m, and the interval between the bottommost hydrophone and the tank bottom is 1 m.

**Figure 3 sensors-22-02420-f003:**
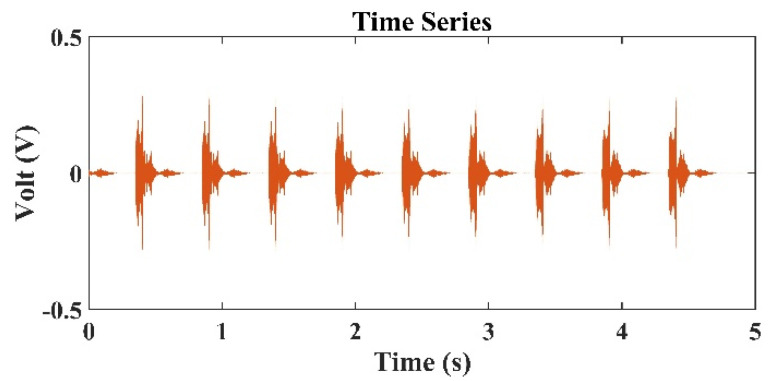
Typical time series of signals received by the second hydrophone ($h_{2}=1.5\;\;m$) in the TRM, which is 80 m from the sound source of 3 kHz.

**Figure 4 sensors-22-02420-f004:**
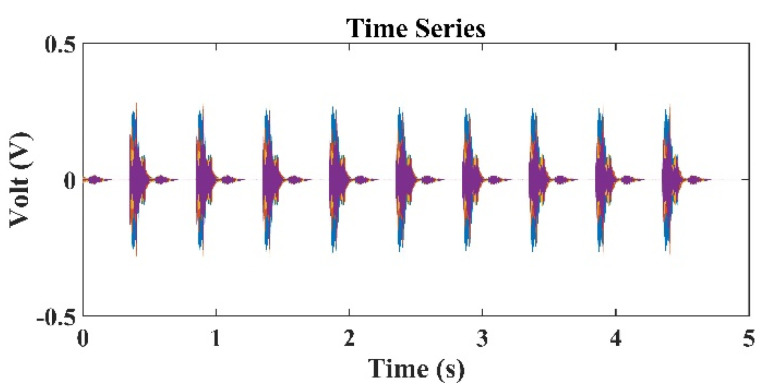
Typical time series of signals received by all hydrophones in the TRM, which is 80 m from the sound source.

**Figure 5 sensors-22-02420-f005:**
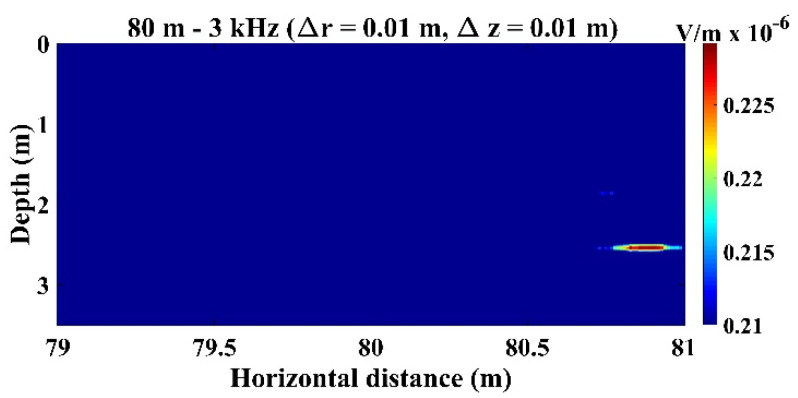
Sound pressure at the retrofocused location in the domain near the sound source, which locates at $r_{o}=80\;m$ and $z_{o}=2.75\;m$. The result indicates that the estimated source location was $r_{oe}=80.89\;m$ and $z_{oe}=2.54\;m$.

**Figure 6 sensors-22-02420-f006:**
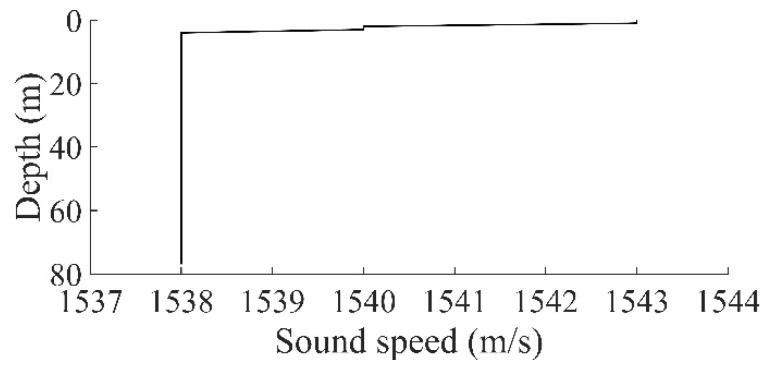
Sound velocity profile at the test location in the offshore region off Small Liuqiu Island.

**Figure 7 sensors-22-02420-f007:**
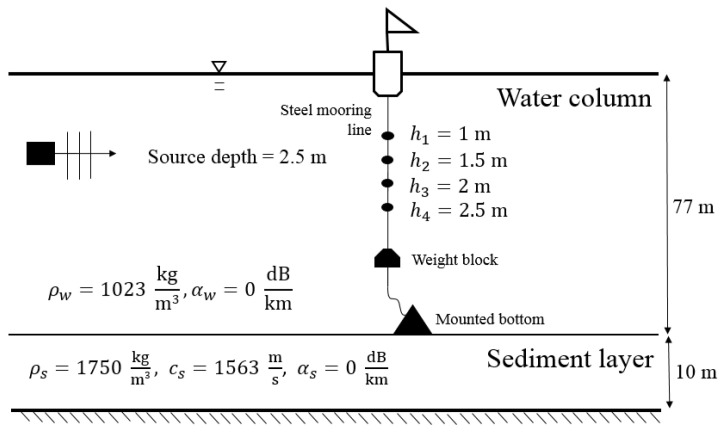
Parameters and geometry of the TRM experiments conducted in the offshore region off Small Liuqiu Island.

**Figure 8 sensors-22-02420-f008:**
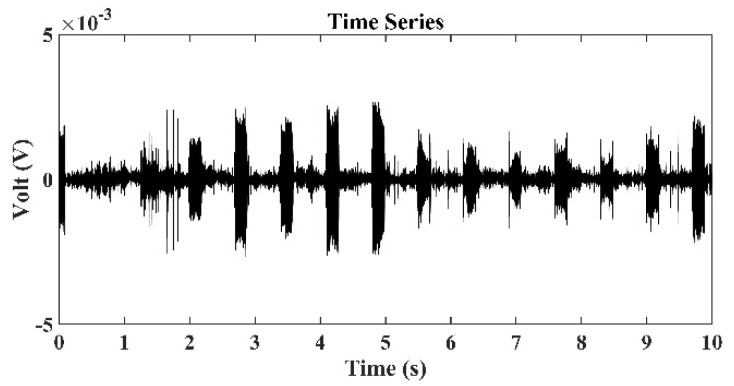
Time series of signals received by the third hydrophone in the TRM, which was 550 m from the sound source with a frequency of 3 kHz.

**Figure 9 sensors-22-02420-f009:**
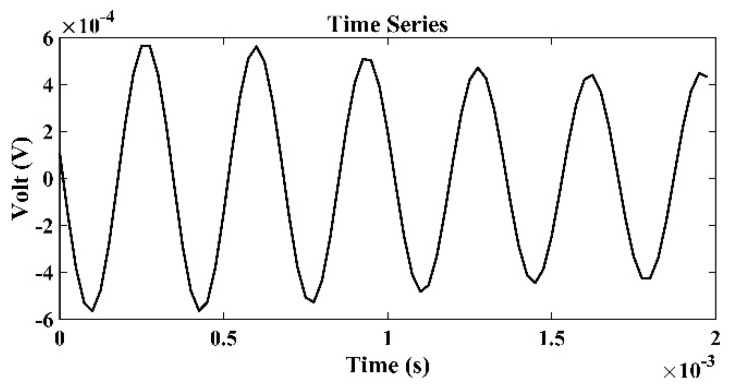
Time series of sound signals within $2\times 10^{-3}\;s$ that were received by the third hydrophone in the TRM, which was 550 m from the sound source with a frequency of 3 kHz.

**Figure 10 sensors-22-02420-f010:**
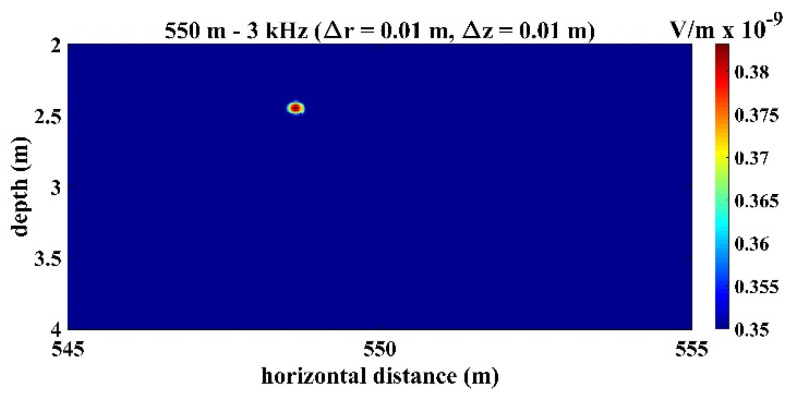
Sound pressure at the retrofocused location in the domain near the sound source, which locates at $r_{o}=550\;m$ and $z_{o}=2.5\;m$. The result indicates that the estimated source location was $r_{oe}=548.6\;m$ and $z_{oe}=2.45\;m$.

**Figure 11 sensors-22-02420-f011:**
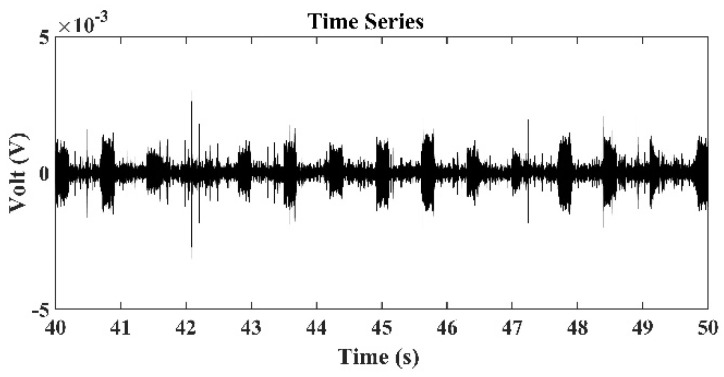
Time series of signals received by the third hydrophone in the TRM, which was 1.6 km from the sound source with a frequency of 3 kHz.

**Figure 12 sensors-22-02420-f012:**
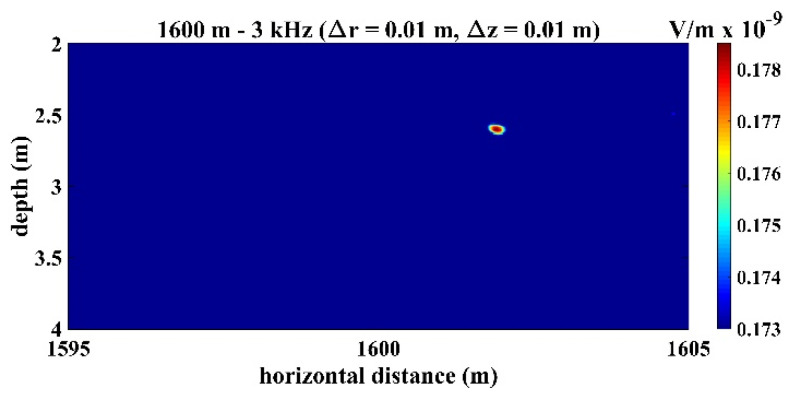
Sound pressure at the retrofocused location in the domain near the sound source, which locates at $r_{o}=1600\;m$ and $z_{o}=2.5\;m$. The result indicates that the estimated source location was $r_{oe}=1601.9\;m$ and $z_{oe}=2.60\;m$.

**Table 1 sensors-22-02420-t001:** Estimated sound source locations at various source locations and sound frequencies obtained from the TRM experiments conducted in the towing tank.

Actual Source Location	$\mathrm{Estimated}\;\mathrm{Sound}\;\mathrm{Source}\;\mathrm{Location}\;(r_{oe},\;z_{oe}$)					Average Errors
	3 kHz	4 kHz	5 kHz	6 kHz	7 kHz	
$r_{o}=4\;m$	3.11	3.00	3.00	3.00	3.00	0.978 m (24.45%)
$z_{o}=2.75\;m$	3.22	2.84	2.85	2.22	1.47	0.494 m (17.96%)
$d_{01}$ (m)	1.01	1.00	1.00	1.14	1.62	1.154 m
$r_{o}=10\;m$	9.25	9.08	9.18	9.06	9.20	0.846 m (8.46%)
$z_{o}=2.75\;m$	2.47	1.10	1.46	0.74	1.05	1.386 m (50.4%)
$d_{01}$ (m)	0.80	1.89	1.53	2.22	1.88	1.664 m
$r_{o}=20\;m$	19.00	19.39	19.02	19.41	19.50	0.736 m (3.68%)
$z_{o}=2.75\;m$	2.55	1.94	3.00	1.82	0.58	0.872 m (31.71%)
$d_{01}$ (m)	1.02	1.01	1.01	1.10	2.23	1.274 m
$r_{o}=40\;m$	39.33	39.00	40.22	39.01	39.78	0.732 m (1.83%)
$z_{o}=2.75\;m$	2.53	2.48	2.95	1.05	0.83	0.862 m (31.35%)
$d_{01}$ (m)	0.71	1.04	0.30	1.97	1.93	1.190 m
$r_{o}=60\;m$	59.26	59.00	60.27	59.05	59.81	0.630 m (1.05%)
$z_{o}=2.75\;m$	0.48	2.52	2.06	1.75	0.83	1.222 m (44.44%)
$d_{01}$ (m)	2.39	1.03	0.74	1.38	1.93	1.494 m
$r_{o}=80\;m$	80.89	80.71	79.04	80.92	80.49	0.794 m (0.99%)
$z_{o}=2.75\;m$	2.54	2.29	1.74	2.48	1.29	0.682 m (24.80%)
$d_{01}$ (m)	0.91	0.85	1.39	0.96	1.54	1.130 m

$d_{01}$: distance between the actual and estimated sound source locations.

**Table 2 sensors-22-02420-t002:** Estimated sound source locations at various source locations and sound frequencies obtained from the TRM experiments conducted in the offshore region off Small Liuqiu Island.

Actual Source Location (Range, Depth)	$\mathrm{Estimated}\;\mathrm{Source}\;\mathrm{Location}\;(r_{oe},\;z_{oe}$)					Average Errors
	3 kHz	4 kHz	5 kHz	6 kHz	7 kHz	
$r_{o}=550\;m$	548.6	548.4	552.8	551.3	548.5	1.72 m (0.31%)
$z_{o}=2.5\;m$	2.45	2.87	3.09	3.36	2.04	0.466 m (18.64%)
$d_{01}$(m)	1.36	1.66	2.89	1.58	1.55	1.808 m
$r_{o}=1100\;m$	1095.0	1102.9	1100.6	1102.4	1097.7	2.64 m (0.24%)
$z_{o}=2.5\;m$	3.21	4.00	2.57	2.65	2.51	0.468 m (18.72%)
$d_{01}$(m)	5.05	3.26	0.60	2.40	2.30	2.722 m
$r_{o}=1600\;m$	1601.9	1600.2	1600.2	1600.5	1595.3	0.70 m (0.044%)
$z_{o}=2.5\;m$	2.60	3.44	3.44	2.84	3.89	0.742 m (29.68%)
$d_{01}$(m)	1.90	0.96	0.96	0.60	4.90	1.864 m

$d_{01}$: distance between the actual and estimated sound source locations.

## Data Availability

Not applicable.

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
