# Peer review of "Underwater Sound Source Localization Based on Passive Time-Reversal Mirror and Ray Theory"

_sensors, 2022, doi:10.3390/s22062420_

Round 1

Reviewer 1 Report

The authors present an experimental study on sensing the localization of underwater acoustic sources. This topic falls into the journal’s scope. The paper is unusually detailed. If the authors can satisfactorily address my below-listed concerns and questions, I am happy to support publication of this manuscript in this journal.

  1. The authors make some vague statements about the role of absorption in time reversal (TR). TR only works perfectly in systems without any absorption, i.e. not in any realistic experiment. The role of absorption is not clearly discussed in the introduction and section II on TRM theory. Where does absorption come into the Equations of Sec. II ? This needs to be reworked.
  2. The above point leads on to a second question : why are the results so good despite imperfect TR due to absorption ? Indeed, even in systems with strong absorption like microwave cavities, TR works quite well. Maybe this question is beyond the authors’ knowledge and paper scope though, so this is not a mandatory question.
  3. The authors just say that TR and phase conjugation are related, but they do not clearly state explocitly that TR is broadband time-coherent phase conjugation. This must be better explained for the reader.
  4. The authors list various localization techniques, but they miss wave-fingerprint (WFP) based techniques. Localization through synthetic TR is just a case of WFP in which TR analysis is used. The authors should clarify this and contextualize the broader class of wave fingerprinting. WFP can use temporal degrees of freedom as in TR, but it can also use spatial or configurational DoF where available. See for instance https://doi.org/10.1103/PhysRevLett.121.063901
  5. A key advantage of WFP in the above example is that it works despite not having analytical descriptions of the highly complex ray paths. Yet in their paper, the authors combine TR with ray tracing. What is the utility of TR compared to simple DOA in settings that are simple enough that one can perfom analytical ray tracing ?
  6. The authors’ indoor experiment with a tank is conceptually very different from what they assume because the tank is a cavity, see for example https://doi.org/10.1121/1.1369101 Indeed, the displayed impulse responses of the authors show clear reverberation. How is this compatible with the assumtion of free space ? are the authors tacitly applying time gating ?
  7. To what extent is the ocean like free space ? what about reflections at surface, bottom, different salinity layers, boats ?
  8. It is also important to understand that deviations from free space that induce multipath make life more complicated but they also improve the localization precision, see https://doi.org/10.1103/PhysRevLett.127.043903 because these results are generic to all wave phenomena, i wonder how they may manifest in the authors’ problem.

Reviewer 2 Report

The manuscript reported an interesting work on a passive sonar array system for detecting the sound wave source using ray theory location in underwater conditions. The work included theory descriptions and experiments in pool and sea. The posted results showed exceptional performance on the proposed method. However, in the manuscript, the authors have not clear novelty statement and discussions. Hence, the reviewer suggested a revision on the manuscript to address the following comments.

  1. The novelty of this work needs to be clearly stated. The reviewer believes that Ray theory is one of the most common ways in underwater detection. And passive receiver array is also used in many studies. The difference between the current work and the existing methods needs to be explained in detail.
  2. The performance presentation needs be enhanced. The reviewer suggests showing the accuracy in the scale of numbers of the operating wavelength. Once the performance was normalized into wavelength scale, the performance of the proposed method is able to compare with the sonars in literature as a part of the discussion.
  3. Additional discussion is needed before the conclusion part. As the reviewer knows the ray theory only is a linear propagation theory which not considered any diffraction, dispersion, or scattering. In the ocean, those facts are better to be taken into account. The reviewer suggested the authors add a discussion regarding to that.
  4. The formatting of the manuscript was not fully according to the template. The equations’ font styles and reference formats were not corrected.

Round 2

Reviewer 1 Report

I am satisfied with the revisions.

Reviewer 2 Report

After the revision, the reviewer agreed that the current version of the manuscript was decent enough. The reviewer recommends an acceptance.